# Visual art inspired by climate change—An analysis of audience reactions to 37 artworks presented during 21st UN climate summit in Paris

**Christian Andreas Klöckner**⊙*⊙, **Laura K. Sommer**⊙⊙

Department of Psychology, Norwegian University of Science and Technology, Trondheim, Norway

⊙ These authors contributed equally to this work.
* Christian.Klockner@ntnu.no

**Data Availability Statement:** The data is available in the repository of the Norwegian Center for Research Data: https://search.nsd.no/study/NSD2630?record-type[0]=study.

## Abstract

This paper suggests and tests a psychological model of environmental art perception and subsequent support for climate change policies. The model is based on findings from art perception and environmental psychology, which indicate that the response of the viewer to the artwork is (1) first an emotional reaction, which can be positive and/or negative. The emotional activation leads to (2) evaluation of the perceived quality of the artwork. This forms the first impression of the artwork the viewer gets, which then triggers (3) reflections on the artwork that are finally related to support for climate policies. The model test uses data collected at the ArtCOP21 that accompanied the 21st UN climate summit in Paris. At 37 connected events, the research team collected 883 audience responses with a brief quantitative paper-pencil questionnaire. The data were analyzed using a multilevel-structural equation modeling approach. Results support the suggested theoretical model. Moreover, the effect of reflections on the artwork on support for climate policies is moderated by environmental attitudes, meaning the lower the environmental attitudes, the higher the effect of reflections on policy support. Finally, artwork features like color, size, displaying something personal, etc., could be identified that had a significant relation to differences on the artwork level regarding the first impression of the artwork and the reflections elicited. The study shows that being confronted with climate change-related artwork relates at least in the short run to increased climate policy support, which is mostly channeled through an emotional activation with following cognitive processing. Features of the artwork relate to how strongly and which emotions are activated.

## Introduction

Global environmental change, such as climate change, is a topic that engages scientists as well as artists working with many art forms. Artists who respond to such changes provide their personal interpretations of the topic. With respect to climate change, the art scene is evolving quickly [1–3]; however, very little is known about how being confronted with such artwork

**Funding:** This research was funded to CAK by the Norwegian Research Council (www.nfr.no) in the Climart project (project number 235223). The funders had no role in study design, data collection and analysis, decision to publish, or preparation of the manuscript.

**Competing interests:** The authors have declared that no competing interests exist.

affects the audience. Specific features of art, such as its ability to evoke emotions, are said to differentiate this form of communication from standard climate change communication derived, for example, from a scientific report [4, 5]. This could increase engagement in the topic, given that inaction towards climate change and other environmental changes of global concern has been linked to the lack of emotional involvement of the audience [6]. Thus, art may serve as an interesting component in a society's response to global environmental change, functioning as an emotional motor of behavior change. A literature review by Roosen, Klöckner, and Swim [4] compares art to other means of climate change communication. They argue that contemporary environmental art can, for example, disrupt habits and routines, offer the spectator space for reflection, strengthen a sense of group identity among the spectators and thereby overcome psychological barriers for taking action. Based on these initial ideas, this paper studies if art inspired by environmental changes relates positively to the support of climate action in its audience, utilizing a climate change-related art event as a study arena. The paper further delves into exploring the potential mechanisms behind the effects art might have and analyses features of the artworks that resonate particularly well with the psychological key variables identified.

Empirical studies addressing the psychological link between climate change-related art, audience responses, and behavioral change have shown that art can trigger both positive and negative emotions for the environment [7]. Moreover, art was found to facilitate communication between stakeholders [7–10] and to increase engagement in group discussions [9, 11]. One study included a large selection of artworks, which allows examination of features of *different* artworks and their impact on the audience [—ANONYMIZED—]. Utilizing a large international climate change art festival in Paris (ARTCOP21), which was organized in parallel to the political negotiations of COP21, the study investigated psychological responses to 37 different artworks (including a large variation of styles, art forms, and presentation venues).—ANONYMIZED—found that being confronted with certain environmental artworks correlated to different emotional reactions, grouped them accordingly in clusters, and investigated the thoughts participants had when seeing the artworks. Furthermore, they identified characteristics of artworks in the clusters to relate them to the emotional and cognitive reactions that went along with experiencing the artworks in the clusters. The researchers conclude that colorful artworks, which make the cause and effect of human behavior visible while offering solutions and showing sublime nature related to the strongest emotional and cognitive reactions relative to the artworks in the other clusters.

The present study is based on the same sample and data but takes a very different analysis perspective. Whereas the first study focused primarily on artwork features, the present study aims to go beyond the relative description of emotional reactions between clusters of artworks and dig deeper into the psychological mechanism of processing the confrontation with climate change inspired artwork on the level of individual audience members. Using Multilevel Structural Equation Modelling, we research relations between the psychological reactions, art, and its potential impact on spectators. We first outline the context for our work by describing where we see the potential for environmental art in climate change communication. Next, we suggest a model, which incorporates the psychological variables and the way we expect them to influence pro-environmental behavior in the form of policy support. Afterward, we test the model and finally discuss the results and their implications.

## Exploring the psychological potential of climate change-related art

An art experience is often described as creating a special connection to its viewer through conveying meaning, eliciting pleasure, or by offering a manner of communication [12]. Pelowski,

Markey, Lauring, and Leder [13] visualized the mechanism of art perception and its effect by summarizing models that have been suggested in the field of empirical aesthetics in the last 20 years. These influential models [14–20] describe the psychology of perceiving art as a process involving input, processing, and output stages. We considered especially the input and the output stage as relevant for the present study. In the following sections, we will explain why we focus on these two stages and outline which input and output variables we considered relevant for the environmental art perception model we suggest. The model also incorporates findings from environmental psychology on what factors are essential to motivate people to change.

**Input variables relevant in the climate change art experience.** The input stage is a stage in which different aspects, such as features of the artwork as well as features of the viewer and the surrounding, flow into the mechanism of art perception. The input variables included in this study will be described in detail below.

*Characteristics of the viewer*. Socio-cultural factors are important characteristics of the spectator and relevant input factors in the process of art perception and environmental studies. For one, the more art experience a person has, the stronger the emotional reactions, as well as understanding and appreciation of an artwork [21]. People with more art experience were also found to be more flexible in their art perception, while for people with less experience, the connection between their understanding and appreciation of the artwork was much more correlated [22]. This means that novices appreciate art much more if they understand it, which is also supported by Leder, Gerger, Brieber, and Schwarz [23], who found that experience with art reduced negative emotional responses to provocative or negative art.

In the environmental domain, younger people, women, and people with a higher level of education have been reported to be more environmentally concerned than older people, men, or people that are less educated [24–27]. This leads us to assume that characteristics of the viewer, such as art expertise, age, education, and gender, would affect the processing and output of art perception, and we controlled for these aspects in our analyses.

Environmental attitude is a tenacious psychological factor, which we assumed not likely to change through one encounter with environmental art [28, 29] but to possibly influence the perception of environmental art. It is one of the main variables of the theory of planned behavior affecting behavior mediated through intentions [30] and was identified as one of the most powerful predictors of environmental intention and subsequent behavior [31].

*Artwork related features*. It is likely that the reaction to climate change-related art not only depends on the characteristics of the observer but also of the artwork. Therefore, we documented basic features of the artwork such as the form of visual art (e.g., photography, painting, sculpture, etc.), its size, and materials used in its creation, as well as indicators for neuropsychological principles, which were found to be responsible for why watching art is considered a stimulating and pleasant experience by most people [32], and thus might also be decisive for the effect the artwork might achieve:

1. The peak shift principle, which is an exaggeration of key defining features of a form or object. An object which is defined by being long might be displayed as even longer in an artwork, or an object that is colored in a warm yellow tone against a dark background will be colored even more intensively against an even more dark background.

2. Perceptual grouping and binding are Gestalt principles [33] and describe the effect that similar items (similar shape, color, movement, etc.) are automatically grouped to larger objects.

3. Isolation of single visual cues, which refers to the effect that artworks often reduce the number of visual stimuli to a small selection, which then activates the respective neuronal clusters more intensively.

4. Problem-solving, which is an effect of the ambiguity of artworks that offer several alternative ways of perception and interpretation and thus engage the brain in this problem-solving activity.

5. Contrast extraction: In addition to the peak shift principle, the inclusion of strong contrasts in artworks might be stimulating for the brain since it has developed for extracting meaning from such contrasts.

6. Symmetry: As opposed to contrast and peak shift, symmetry can be a stimulating and pleasing feature of artworks because it provides the visual system with stability.

7. Use of metaphor: Finally, artwork often makes use of metaphors, which stimulate the observer by telling a new story in a known way.

In addition to these principles, we expected that if the artwork offers an element of participation for the viewers, this will allow them to become active creators of the work and thereby increase engagement with the topic of the artwork [4, 5, 34].

**The processing stage of the climate change art experience.** In the processing stage, early and intermediate processing mechanisms of the artwork take place [15, 35]. Research on this stage is dominated by neuro-aesthetic approaches connecting the processing mechanisms with brain functioning. This is based on the idea that especially early processing mechanisms are bottom-up, immediate, and automated processes and do not underly cognitive control [13, 35]. For methodological reasons (we studied the artworks in the field with short contact time to the respondents), we were not able to investigate the processing stage in the present study but refer the interested reader to Chatterjee's proposed model [14, 36, 37] and research on the processing stage [38, 39].

**Expected output variables of the climate change art experience.** The output stage of art perception is where the focus of research in the field is located [13]. This is also true for the present study since we were mostly interested in the cognitive and behavioral outputs of the climate change art perception process. Major output factors of the art perception process, are according to Pelowski et al. [13], emotional reactions and meaning-making, which can be understood as part of the cognitive output of the art experience. In the following, we will present the output factors we expected would be triggered by the art experience during the ArtCOP21.

*Emotional response to art as a motor of behavior change.* Emotions are substantial for decision making because they fulfill a relevance and a commitment function and motivate people to act [40]. In the context of environmental art, the relevance function focuses attention on specific features of an artwork that are of personal relevance for the viewer. The result is an evaluation in the form of a discrete emotion such as disappointment or hope, which constitutes an appraisal, implying a tendency to act, while the commitment function facilitates social collaboration against the short-term self-interest of people by making people stick to their decisions. Guilt is a typical example of emotion eliciting commitment, and it has been used to explain pro-environmental intentions and climate change mitigating behaviors [41, 42].

Other relevant emotions in the context of climate change and art are fear or anxiety, which can at times block constructive engagement with climate change [43]. According to Pfister and Böhm [40], they might have evolved under evolutionary selection pressure, as part of a fast avoidance response. On the opposite spectrum of emotions, hope [41] and a sense of awe [3] have been associated with climate change mitigation.

For our model, we decided to use a relatively simple categorization of emotions into positive emotions, such as hope, inspiration, etc., and negative emotions, such as anger, fear, etc. We acknowledge that there are differences within these categories, but in terms of model

parsimony, we keep this simple grouping (which is also supported by the factor analytical approach described below). Thereby we follow the tradition in empirical aesthetics to conceptualize the affective reactions in a dichotomous way [44, 45] but adapt it to the environmental setting, beyond 'liking' or 'disliking' an artwork.

*Cognitive response to art as a motor of behavior change.* The process of art perception also includes cognitions, which revolve around finding meaning and identifying a personal connection to the artwork. These reflections can include different aspects concerning the artwork and its topic, as well as the spectator and their understanding of themselves.

Part of the first cognitive processing of the artwork is the forming of a 'first impression of the artwork,' which includes a quick evaluation of the content of the artwork, resulting in a 'perceived quality of the artwork' and an 'identification with the artist' [46]. The 'perceived quality of the artwork' is formed by activating memories and connecting them with the composition, global structural organization, and semantics of the art [13], while the 'identification with the artist' represents an implicit judgment on what kind of person the artist might be. A study by White, Kaufman, and Riggs [47] showed that the background of the artist influences the way the artwork is being perceived.

After the first impression, higher cognitive processes in the perception of environmental art take place, which is summarized as 'reflections and cognitions on the art' in the most influential models in empirical aesthetics [13–20, 44, 48]. These reflections are responsible for finding meaning in the artwork, which may lead to self-discovery [49] and self-improvement [13, 15]. The underlying idea is that the more reflection and the more meaning spectators are able to ascribe to the art, the more they are likely to be motivated to act. In the case of climate change, this can be represented by an increase in support for environmental policies.

We suggest that the general reflections on the art include a 'relevance of climate change in the daily life' of the spectator, an 'awareness of consequences' of one's behavior. Environmental psychology describes an awareness of consequences as becoming aware of negative consequences of human activity on the environment, which was found to facilitate pro-environmental behavior [50–52]. Moreover, awareness of consequences was increased after seeing environmental art [10, 53]. While a connection to one's daily life is needed to make the topic of climate change personal and to constructively engage people with environmental problems [43]. Same as the connection to people's daily life, the subjective perception of how likely climate change will affect them can be an important cognitive step that makes climate change personal.

**A suggested overall structural model.** The design of the present study allows to bring together characteristics of the audience and the artworks included since the number of artworks and participants per artwork is sufficient for a multilevel structural equation modeling approach. However, the correlational nature of the data does not allow for causal inferences, so this needs to be taken into consideration when interpreting the results presented.

Based on the considerations presented in the previous sections, we expect that the relation between art experiences and climate policy support depends both on person characteristics (e.g., socio-demographics, art expertise, environmental attitudes, etc.) and characteristics of the artwork (e.g., its form, material, the neuro-aesthetic principle applied by the artist, etc.). Fig 1 presents a stylized overview of the analysis framework, which we used to structure the following analyses.

We expect that there is a chain of effects when a member of the audience is confronted with a climate change-related artwork. In the order of the different effects, we follow the model proposed by Locher [46], since other models are less clear on the order of emotional and cognitive reactions [13]: When the viewer encounters an artwork, their perception is influenced by their personal characteristics, as well as characteristics of the individual artwork. The exposure to

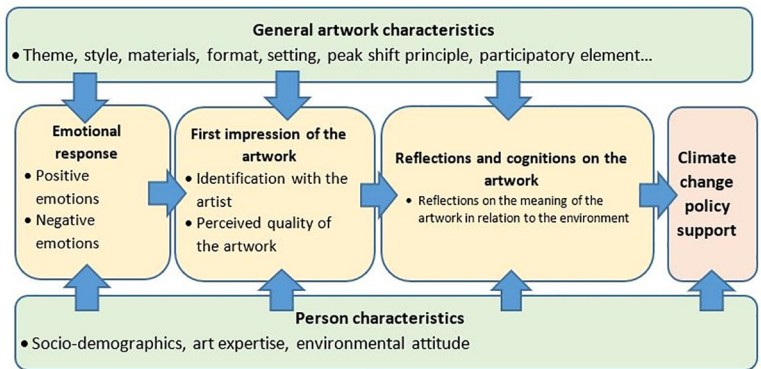

**Fig 1. Analysis framework for the relation between art and policy support.** In green: input variables; in yellow: cognitive and emotional outcome variables; in red: behavioral outcome variable.

the artwork relates to emotions, which can be negative or positive, or both. This is theoretically followed by a spontaneous generation of a first, general impression of the artwork. This impression contains the perceived quality of the artwork and a sense of how much the viewer can identify with the artist. The next step is to satisfy cognitive curiosity and includes general reflections on the artwork and climate change, climate change perception. The cognitions may then eventually relate positively to the motivation to act and to an environmentally friendly choice or action, or support of climate policies. Specific features of an artwork or the individual may make certain steps of this process leading to policy acceptance more or less effective.

## Method

### Data collection procedure

In November 2015 and January 2016, the Climart research team traveled to Paris to survey audience responses to artworks presented in connection to the ARTCOP21 event. ARTCOP21 was a large conglomeration of exhibitions of climate-related art. Between September and December 2015, the ARTCOP21 art event was organized as a "global festival of cultural activity on climate change" (www.artcop21.com) accompanying the climate negotiations in Paris. In total, 551 events in 54 countries were registered in the project database. Ninety-nine of these events were taking place in Paris in November 2015, 50 of them categorized as "visual art," which was the focus of our analysis. Artworks were selected to cover a range of different visual art forms, including paintings, sculptures, video installations, and happenings, and were displayed in established art galleries, churches, a science museum, and at outdoor locations (a list of artworks can be found in S1 Table). The study was approved by the Norwegian Center for Research Data (approval number 45125). Oral consent was given by the participants.

### Sample

In total, 883 individuals from the audiences at 37 different artworks were surveyed with a four-page paper-pencil questionnaire, an average of 24 participants per event (with a range of 15 to 38). This creates a data structure, where responses to the questionnaire are clustered into the different venues where the artworks were presented. The questionnaire was provided in either French or English, depending on the participants' language abilities, and took about 10 minutes to answer. Most of the respondents (78.7%) answered the French version of the questionnaire. Most (69.9%) of the respondents reported having French nationality. Other larger

groups were Spanish, US American, Dutch, English (each 2.0%). Slightly more than half of the respondents (56.2%) were female, 43.8% were male. The mean age of participants was 36.6 years (SD = 16.4). Most (71.0%) of the participants hold a university degree; the remainder held a college or technical degree (9.2%), a high school diploma (15.9%), or a primary school diploma as the highest degree (3.9%). Participants had a strong interest in art, which is not surprising since at least some of the studied pieces were presented in gallery spaces. Involvement in art ranged from about a third (33.1%) describing themselves as art-lovers, half (50.9%) reporting that they like watching art, and some (14.2%) indicating that they sometimes like art, sometimes not. Very few (1.8%) reported that they do not like art at all. However, the percentage of art experienced people in relation to less experienced people varied strongly between venues (for single artworks that were presented in public spaces or a science museum, up to 40% placed themselves in the "sometimes like/sometimes do not like art" or "do not like art" categories). Only some (9.6%) reported that they knew the artist who produced the respective piece from before.

## Measures

The survey instrument was designed to capture the variables outlined in the theoretical model above. The items were developed specifically for this study by the research team, based on the literature presented in the introduction. All items were treated as ordered-categorical in the analyses, which results in that the links between the latent variables and the items in the models become probit links instead of linear regression links. A confirmatory factor analysis was conducted based on the expected variable structure and received a good model fit ($Chi^2$ = 452.12, df = 128, p < .001, $Chi^2$/df = 3.53; RMSEA = .054 [.048 .059]; CFI = .96; TLI = .95; SRMR = .043). See S2 and S3 Tables for the loadings of the indicators on the respective latent variables, correlations between the variables, and measures of composite reliability and discriminant validity. In general, the deeper analysis of the measurement model underlines its good fit as the composite reliabilities of three out of four latent variables are strong, also corresponding to high Average Variance Extracted (AVE) scores. However, for the "identification with the artist" variable, the numbers indicate a challenge with the internal consistency of the measure and potential overlap with the "reflection on the artwork" variable. Therefore, we decided to drop the variable "Identification with the artist" from the analyses. Removing the latent variable and the corresponding indicators from the measurement model has little effect on the overall model fit (however, RMSEA and the $Chi^2$/df indicate a slightly worse fit of the model without "identification with the artist": $Chi^2$ = 424.17, df = 87, p < .001, $Chi^2$/df = 4.88; RMSEA = .066 [.060 .073]; CFI = .96; TLI = .94; SRMR = .044). The lower part of the S2 Table and the above diagonal part of the S3 Table display the estimates for the measurement model with "identification with the artist" excluded.

**The emotional reaction to the artwork.** The participants were asked to what extent the artwork brought up each of a list of feelings within them. From this list, happiness, hope, inspiration/enthusiasm were defined as positive emotions, whereas anger, anxiety, and sadness/disappointment that nothing is happening to prevent climate change, were defined as negative emotions. The items were answered on a seven-point scale (1 = not at all, 7 = very much).

**Perceived quality of the artwork.** The perceived quality of the artwork was measured with one item: "The artwork appears to be of considerable artistic quality." The item was answered on a seven-point scale (1 = strongly disagree, 7 = strongly agree).

**Identification with the artist.** Identification with the artist was measured by three items. This variable was planned to be included in the analysis as it is part of the conceptual model but was dropped from the analysis due to poor reliability and discriminant validity (see

above). The participants were asked if they imagined the artist to be (1) someone like themselves, (2) someone with values similar to themselves, and (3) someone expressing the values of the public. As indicated above, these items were apparently measuring relatively different components, so conclusions with respect to the role of identification with the artist should be drawn with caution. The items were answered on a seven-point scale (1 = strongly disagree, 7 = strongly agree).

**Reflections on the artwork.** This variable consists of six components capturing the degree to which the artwork made the audience reflect on climate change: (1) The artwork makes me think and reflect on its meaning, (2) the artwork seems relevant to my daily life, (3) the artwork highlights consequences of climate change that would affect me personally, (4) the artwork makes me think about the problem of climate change, (5) the artwork makes me think about my own role within the current climate situation, and (6) the artwork makes me more aware of my behavior's impact on the environment. The items were answered on a seven-point scale (1 = strongly disagree, 7 = strongly agree).

**Support for climate policy.** As dependent variable, we decided to use the priority of climate policies expressed in a longer block on policy priorities as an answer to the item: How important is it to you that climate change and the environment are given a high priority in policymaking? The answers were (1) it is rather unimportant in comparison, (2) it is not that important to me, (3) it is important but not the highest on the list, and (4) it should have the highest priority. This item was presented in a block of different policy priorities (e.g., economic prosperity, security, etc.) to give it a context. The other policy priorities are not analyzed further in this paper.

**Control variables.** To control for confounding influences of different compositions of audiences, the following variables were included in all analyses: Age, gender (0 = female, 1 = male), environmental attitude (measured by the item: "Comparing yourself to others, how interested would you say that you are in environmental issues?" 1 = far below average; 4 = about average; 7 = far above average), and experience with art (as introduced above in the description of the sample). In the initial analyses, age and experience with art did not relate to the model variables, so they were excluded from further analyses.

**Artwork characteristics.** For each artwork, one to two members of the research team recorded characteristics describing the artwork, for example, the type of artwork (painting, sculpture, photograph, etc.), the materials used, the size, if it was an artwork with participatory elements if it depicted familiar objects, animals, humans, etc. Also, basic features of art perception psychology were captured (e.g., if the artwork made use of the peak shift principle). These characteristics were used in the artwork level analyses described below.

## Analysis strategy

The data has a nested two-level structure since the audience responses have been recorded at 37 venues for artworks in Paris, with a number of people per artwork observing the same artwork. This allows for simultaneous tests of effects of the artwork (e.g., how features of the artwork impact the emotional response it evokes) and effects of person-level factors (e.g., the link between the strength of an individual emotional response and perceived quality). SEM software MPLUS 8.3 was used for the analyses [54]. The COMPLEX analysis type in MPLUS was used, adjusting standard errors for effects of clustered / nested data.

Since the research question of this study taps into new ground, we follow a partly confirmatory (step 1), partly exploratory strategy in the data analysis (step 2 and 3):

1. After the initial confirmatory factor analysis (see supporting information), we ran a confirmatory model test on the structural model as has been derived theoretically in the

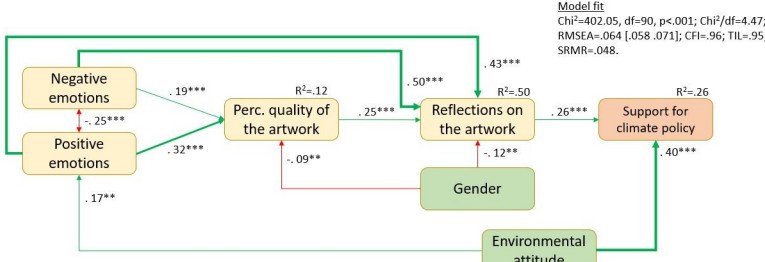

**Fig 2. Person-level model of an art experience's influence on support for climate policy (N = 839).**

introduction on the person level, adjusting the standard errors of the analysis for the clustered data structure. While this first analysis is based on a latent variable model for emotions and reflection on the artwork, the following steps of the analysis could not have been conducted with latent variables due to high model complexity. Therefore, a latent variable score was assessed for each participant for the latent variables in the model estimation process, which was then used in all further analyses. As the latent variable score was assessed during model estimation (and not calculated as a mean or sum score afterward), the latent variable scores are regarded as free from measurement error. The results of step 1 are displayed in Fig 2.

2. In the next step, we tested an interaction between the environmental attitude and the reflection on the artwork on the support for climate policy to find out if the reflections related to the art experience had the same strength for people with varying levels of environmental attitude. This analysis was conducted based on the assumption that people with strong environmental attitudes to start with might support climate policies anyway, whereas people with weaker environmental attitudes might be more affected by the art experience since their policy support is initially lower. To reduce model complexity, only the three relevant variables were included in the analysis (plus gender as a control variable). The results of step 2 are reported in Fig 3 (gender is omitted from the figure since it did not have a significant effect).

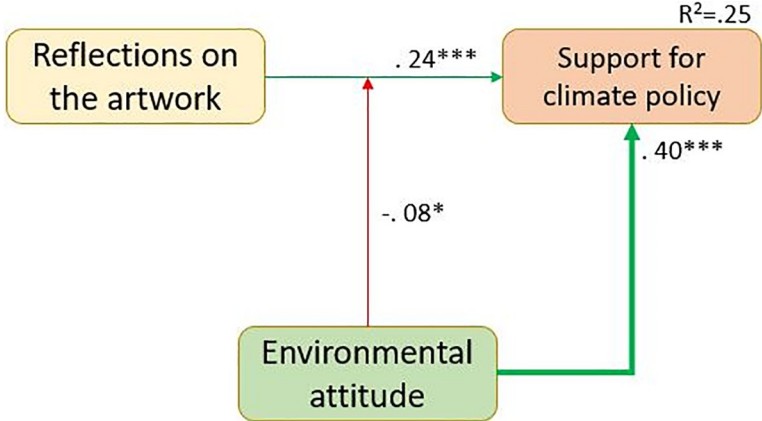

**Fig 3. Person-level model of the interaction between reflection on the artwork and environmental attitudes (N = 761).**

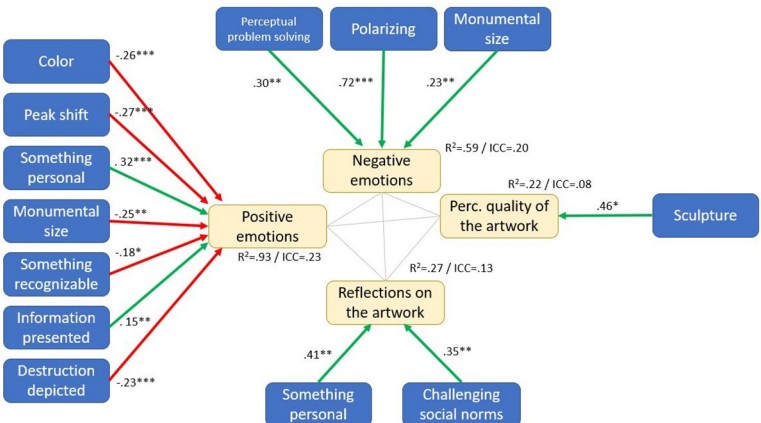

**Fig 4. Artwork level part of the multilevel model with variance in emotional responses, identification with the artist, perceived quality of the artwork, and reflection on the artwork modeled on the artwork level ($N_{level1}$ = 714, $N_{level2}$ = 30).**

3. In the third step, the research focus was on the characteristics of the artworks. The question is if the variables identified as relevant in the first step on the person level differed between artworks. To achieve this, the model tested in step 1 was reconfigured as a full two-level model. All dependent variables in the layers "climate change-related cognitions" and "art experience" were tested for significant variance on the artwork level and all variables except the support for climate policies had significant variation between artworks. The artwork-level variance was then regressed on the artwork features. Since a large selection of characteristics was recorded, this was done in a stepwise procedure, where each level 2 variable was first regressed on all artwork features and then, one-by-one, features with the highest p-level were removed until only significant influences remained. We are aware that this is a highly exploratory strategy, very vulnerable for capitalizing on chance. The results of this analysis should thus be treated as a first indication only. The results of this analysis are reported in Fig 4.

## Results

In this section, we first report the final structural model of the relations between the variables and behavior, then the test of the interaction hypothesis, and finally, the test of a full two-level model with influences of artwork characteristics.

### How does an art experience relate to support for climate policy?

Fig 2 displays the results of the model test. All relations are confirmed, and the model receives a good model fit (see numbers in the figure). The displayed estimates on the arrows are standardized regression weights.

According to the tested model, the first step in the process is the emotional activation that is related to experiencing an artwork. Both positive and negative emotional reactions then relate significantly to the mediating cognitive variables perceived quality and reflections on the artwork. The relation of negative emotions with reflections on the artwork seems to be stronger than the relation of positive emotions, whereas the perceived quality of the artwork is slightly stronger related to positive emotions. The more negative and positive emotions are elicited, the higher is the perceived quality, and the stronger are the reflections on the artwork. The

stronger the reflections on the artwork, the more support for climate policy is reported. Men reported less reflection about the artwork and lower perceived quality of the artworks, whereas environmental attitudes were strongly and positively related to support for climate policies, but also to positive emotions.

## Does the relation between reflection on the artwork and support for climate policy depend on a person's environmental attitude?

To test the assumption that the induced reflection about the artwork might have a stronger relation to policy support depending on differences in environmental attitudes, we tested a model where support for climate policies was regressed on reflection about the artwork, environmental attitudes, and the interaction term between the two (controlled for gender). For the calculation of the interaction term, the variables were grand mean-centered first to avoid false multicollinearity. Fig 3 displays the results of this analysis. The exploration of the interaction term indicated a negative interaction between the two variables, showing that the relation between reflections on the artwork and support for climate policies is stronger for people with weaker environmental attitudes than for people with stronger attitudes. See S4 Table for all model parameters.

## Do different artworks relate to different states in the psychological variables?–A two-level model

In the next step, the same model as in Fig 2 was specified, but this time as a two-level model. Explicitly modelling level 2 variance in dependent variables in the model on the artwork level (which means for all people answering at the same artwork). This allowed examination of whether different artworks elicited systematically different responses among their visitors. For all variables but the support for climate policies, there was meaningful variance among artworks. Then a simplified model was constructed where only the relevant variables on the person level were included and allowed to correlate. This was done to reduce model complexity for the following analyses. Afterward, we examined the artwork features that related significantly to the five psychological variables by testing all recorded features stepwise. The model presented in Fig 4 is the result of this endeavor.

The first result that can be extracted from Fig 4 is that the emotional responses have the most variation between the artworks (the Intra Class Correlations, ICC, indicate that 23% of the variance in positive emotions and 20% of the variance in negative emotions are between artworks). For the other two variables, the variance between artworks is lower.

The variance between artworks in positive emotions can almost completely be explained by them being colorful or black/white (black and white artworks were related to less positive emotions), if they made use of the peak shift principle (if yes, less positive emotions were reported), if something personal to identify with was displayed (which related to more positive emotions), if they were large in size (which was negatively related to the amount of positive emotions), if they depicted known objects (which was also related to less positive emotions), if they presented information about the artwork (which related to more positive emotions), and finally if they depicted destruction (not surprisingly related to less positive emotions).

The variation between artworks in negative emotions triggered can be explained by their size (the bigger, the more negative emotions), if they were polarizing in their design (if yes, then negative emotions were more likely), and if they triggered perceptual problem solving (which also enhanced negative emotional reactions).

Sculptures, as opposed to other types of artworks, were perceived as of higher quality. The level of reflection on the artwork was higher for artworks that challenged social norms and that included something personal to identify with.

## Discussion

This paper reports results from analyses conducted in a unique dataset collecting audience responses to 37 different climate change-related artworks and art events displayed during one big climate change art event coinciding with the climate summit in Paris. The study was conducted to provide more indications for how climate change inspired art relates to psychological responses in its audience and by which mechanisms on the individual level these responses might be explained. Furthermore, the design of the study with surveying audiences at a large number of artworks allowed to combine this with an analysis of how artwork differences affect the person-level psychological variables.

While acknowledging the exploratory and correlational nature of this research, the analysis found that the emotional reactions to the artworks, both negative and positive, were indirectly related to more climate policy support, mediated through reflections on the artwork. Furthermore, we demonstrate not just the role emotions may play in climate change communication through art but also define what characteristics of an artwork may make it more impactful. It is, for example, highly remarkable that positive emotions and reflections on the artwork are positively related to that the art highlights something personal to identify with. This supports our understanding that what makes environmental art impactful is the emotional and personal connection it can create to a topic that is abstract and distant for many people [55, 56]. As was described by Gabrys and Yusoff [57], the arts possibly open up a space in which a freer encounter with scientific findings is possible. This is in alignment with findings by Jacobs et al. [58], who describe in their case study on a climate change art exhibition that it is vital to give people the chance to discover their own meaning within the art and climate change.

In more detail, the analysis found the following relationships between personal responses and climate change art: First, the audience reported emotional reactions to most artworks. The spectrum of emotional responses reported is large and varies from artwork to artwork. Interestingly, it does not seem to matter much for the potential effect if the emotions were positive or negative: both kinds of reactions seem to be positively related to reported policy support. Moreover, art seems to affect people through several pathways: If people perceive the artwork of high quality, this relates to deeper reported reflections about the artwork. When this reflection process is going on, it is positively related to policy support for climate change mitigation, especially in people who do not already show such strong support because of strong environmental attitudes, as indicated by the interaction analysis. Even if this conclusion is based on correlational data in our study, this is in alignment with the general idea that emotions motivate behavior through a commitment and relevance function [40] and findings of Leiserowitz [59], who found that climate change imagery and emotions influence risk perceptions and policy support. We also found support for findings that awareness of consequences can be increased through environmental art [53]. The artworks apparently prompted moments of reflection, questioning one's position in and contribution to a problem. These findings on meaning-making and reflections are consistent with models by Locher [20], Leder [15, 16], and others mentioned by Pelowski and colleagues [13].

With respect to differences between artworks, they were strongest in the emotional reaction in the audiences. Given the large variety of artistic expressions, this is not surprising. Some of the artwork features were particularly related to different emotional reactions, but also different levels of perceived artistic quality and level of reflections triggered. These findings give a

first indication of how features of environmental art affect the audience. More positive emotions were found if the artwork included something personal, was not too big, was presented with an explanation of how to "read" it, but did not include destruction, especially of known landmarks. Negative emotions, on the other hand, are likely triggered by polarizing and monumental artworks. Whereas some of these findings might have been caused by the specific selection of artworks, they nevertheless point at some interesting relations between artwork features and how the artworks are perceived, which artists should be aware of.

## Limitations of the study

Despite its uniqueness, the presented study has some important weaknesses that need to be acknowledged. First, the study's correlational design does not allow drawing conclusions on causes and effects among the studied variables. In connection with its strongly exploratory nature on the artwork level, it has generated several hypotheses for future work, but it is not able to conclusively decide that a certain aspect is the cause of another effect. More controlled studies need to replicate and confirm the presented mechanisms and effects. Furthermore, the measurement quality of the variable "identification with the artist" is sub-optimal, which prevented its use in this study.

Second, the literature on art perception is not definite on the order of emotions and cognitions–Locher [46] and Capo et al. [60] place, for example, the emotional reaction first, as we did it in our model, while Leder [16] suggests an iterative process, in which emotions inform all the other steps in art perception in a feedback loop. To check for iterative feedback of emotions, we would have needed to follow our participants longitudinally through the experience probing them repeatedly about their emotional status and cognitive reflections. This was impossible to implement in the circumstances under which this data was collected. Further research should take into consideration whether emotions might inform environmental art perception through all steps.

Third, many of the studied artworks were presented in designated art spaces such as galleries or open-air areas in connection to museums. This also skewed the sample toward people who describe themselves as art interested. The level of art expertise was found by Leder et al. [23] to influence the level of emotional reaction to "negative" art; even if we could not find an impact of art expertise on the emotional reactions and reflections to the artwork in our study, there is still the possibility that participants with a higher level of art expertise perceived the artworks differently than novices.

Fourth, because of the purpose, venue, and timing of the exhibits, it is likely that the audience was more concerned about climate change than the average Parisian resident or visitor, which may have produced different results from what would have been found in a more representative population sample. Because the number of audience members studied per artwork was not large (around 25 per artwork), more detailed analyses were unfortunately not possible that might differentiate between artwork effects on a dedicated art audience and an audience consisting of people during their everyday lives. The willingness to reflect and think might be smaller for a general population sample. Perhaps the emotional triggers would be lost, or perhaps they would be stronger because the experience would be unexpected.

Finally, even if the focus of our study was not to explain support for climate change policies in detail and we, therefore, did not include a large set of potential predictors of support, it can be argued that policy support is usually strongly impacted by variables such as political preferences and worldviews, which we thus were not able to control for. If these variables were related to how people react to artworks (which they well might be), we might falsely assume that the variables in our model explain the differences in policy support, whereas the

differences, in reality, are caused by background variables not included in the study. However, since we included environmental attitudes, which is a variable often strongly related to world-views and political preferences, and we find significant relations between reflections on the art-work and policy support even when controlling for environmental attitudes, plus a significant interaction. We feel confident that the relations reported in our study are not completely arbitrary.

## Conclusion

This is one of the first studies that systematically analyses the psychological mechanisms by which climate change-related art might relate to people's reflections on climate change and support for climate policies. Thereby, it brings together research on the psychological mechanisms of experiences with art and the environmental psychological theory behind support for climate change mitigation. By covering a large variety of artworks, we were able to explore reactions across different types of artworks. The results indicate that being confronted with climate change art mostly relates to emotional reactions. The activating function of art also seems to be related to deeper reflections and support for action in the form of policies. The inputs defined in art perception models [13] may also mediate the effects of viewing environmental art. The conclusion is that our analyses suggest that if art manages to trigger a strong enough emotional reaction, it might start a process of reflection, which then may lead to support of the action. Interestingly, both negative and positive emotions seem to fuel this support for action. Which emotional profile art triggers depends on different characteristics of the art-work, such as technique or size.

The results indicate that climate change-related art might have a unique communicative function based on opening a space in which people can make the topic of climate change personally relevant to themselves. Subjectivity, emotions, or connecting the individual to the global may be easier achieved through the medium of art if art does not claim to tell or find "the truth" about climate change but bring up topics relevant for society and leave it up to the viewer to find their own truth. The results show that this form of communication can be linked to a response in the audience. The link to environmental behavior should be studied further, possibly with an artwork specifically designed to elicit engagement with climate change.

## Supporting information

**S1 Table. List of artworks st.**
(DOCX)

**S2 Table. Loadings on the latent variables in the measurement model (with "identification with the artist" included in the upper half, with "identification with the artist" excluded in the lower half of the table).** CR = composite reliability; AVE = average variance extracted; B = unstandardized loading; SE = standard error; Beta = standardized loading; RES = unstandardized unexplained residual variance; p = significance level of loading.
(DOCX)

**S3 Table. Correlations of all latent variables and single-item measures in the measurement model (model with "identification with the artist" below the diagonal; model without "identification with the artist" above the diagonal).** The table displays Pearson correlations; *** p < .001, ** p < .01, * p < .05.
(DOCX)

**S4 Table. A simplified model to test the interaction between environmental attitude and reflection on the artwork (with gender as a control variable).** As the outcome variable

"support for climate policy" was ordered-categorical, we report the three thresholds in the table. The outcome variable was linked in the regression through a probit link.
B = unstandardized loading; SE = standard error; Beta = standardized loading; p = significance level.
(DOCX)

## Acknowledgments

We thank the other project partners Janet Swim (Penn State University), Paul Stern (Social and Environmental Research Institute, Northampton), Sam Jury (University of Hertfordshire), David Buckland (Cape Farewell), Martina Zienert & Joachim Borner (KMGNE), Edgar Hertwich (Yale University), Peter Huybers (Harvard University), Laura Coleman (ONCA gallery Brighton), and Liselotte Roosen (NTNU) for their valuable input to the project and comments on earlier versions of this paper. Janet Swim and Paul Stern were particularly involved in discussions around this paper.

## Author Contributions

**Conceptualization:** Christian Andreas Klöckner.

**Data curation:** Laura K. Sommer.

**Formal analysis:** Christian Andreas Klöckner, Laura K. Sommer.

**Funding acquisition:** Christian Andreas Klöckner.

**Investigation:** Christian Andreas Klöckner, Laura K. Sommer.

**Methodology:** Christian Andreas Klöckner.

**Project administration:** Christian Andreas Klöckner, Laura K. Sommer.

**Resources:** Christian Andreas Klöckner.

**Validation:** Christian Andreas Klöckner.

**Visualization:** Christian Andreas Klöckner.

**Writing – original draft:** Christian Andreas Klöckner.

**Writing – review & editing:** Laura K. Sommer.

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
