## [Decision Letter · Decision Letter 0]

14 Dec 2020

PONE-D-20-32925

And once more with feeling - What role can visual art play in motivating people to support climate policy?

PLOS ONE

Dear Dr. Klöckner,

Thank you for submitting your manuscript to PLOS ONE. After careful consideration, we feel that it has merit but does not fully meet PLOS ONE’s publication criteria as it currently stands. Therefore, we invite you to submit a revised version of the manuscript that addresses the points raised during the review process.

Please find below the reviewer's comments as well as those of mine.

We look forward to receiving your revised manuscript.

Kind regards,

Valerio Capraro

Academic Editor

PLOS ONE

2. Please modify the title to ensure that it is meeting PLOS’ guidelines (https://journals.plos.org/plosone/s/submission-guidelines#loc-title). In particular, the title should be "specific, descriptive, concise, and comprehensible to readers outside the field" and in this case it is not informative and specific about your study's scope and methodology.

Additional Editor Comments (if provided):

I have now collected one review from one expert in the field. I was unable to find a second reviewer, but the review I could collect is very detailed and helpful. Therefore, I feel confident in making a decision with only one review. The reviewer likes the paper but suggests several improvements. Therefore, I would like to invite you to revise your manuscript following the reviewer's comments.

I am looking forward for the revision.

Reviewers' comments:

Reviewer's Responses to Questions

**Comments to the Author**

1. Is the manuscript technically sound, and do the data support the conclusions?

Reviewer #1: Partly

2. Has the statistical analysis been performed appropriately and rigorously? 

Reviewer #1: I Don't Know

3. Have the authors made all data underlying the findings in their manuscript fully available?

Reviewer #1: Yes

4. Is the manuscript presented in an intelligible fashion and written in standard English?

Reviewer #1: Yes

5. Review Comments to the Author

Reviewer #1: This is a well-written paper that describes an interesting, novel study about an understudied topic--the relationship between art viewing and various emotional and psychological responses. As this is an understudied area, this study has potential to add to this specific field. However, I have significant questions about some aspects of the statistical analyses (especially the factor analysis). Additionally, many of the claims articulated here go further than merited by the data and rely on several assumptions. While I believe this study does have something to offer in regards to climate-related art viewing, more work needs to be done to clarify the validity of this study and to temper its claims. My specific comments are below:

1. Is the manuscript technically sound, and do the data support the conclusions? Partly.

The factor analysis is unclear and lacking key information. Specifically, the authors refer to Table B that is supposed to show loadings on latent variables. While it shows loading of each item, it is unclear what latent variables each is loading on, as well as each item’s unique loading. It is also unclear what percentage of variability in the CFA model is explained by each latent variable. This is an important aspect of accepting the overall results, as analysis relies on these groupings of items as variables and builds the SEM on these groupings. In particular, I have questions about the “Reflection” variable, which seems like it is being measured by quite different items. It is hard to believe that they all load uniquely onto one latent variable, and I would like to see clearer evidence for that before fully trusting the SEM.

I also have questions about some aspects of the SEM and would like further clarification. For instance, on page 9, the authors say: “The data has a clustered two-level structure, since the audience responses have been recorded at 37 venues for artworks in Paris.” What levels are you clustering the data at? For venue? Or for personal characteristics? What two levels constitute the structure?

Is there any error measurement associated with calculation of latent variable scores for each person? Using scores instead of variables could introduce more error, I would think.

A major problem with this paper is that many conclusions are written with causal language when the statistical analyses do not merit that. For example on page 10, there is a section header that says: “How does an art experience motivate support for climate policy?” Despite the identified relationship between the art experience and support for policy, we cannot know from this study anything about cause. As elaborated below, I believe that there are important variables that actually do impact policy support that are not measured or mentioned by the authors. This causal language comes up at several points, including the Discussion. Although the authors address this point in the Limitations, much of the language used earlier on is misleading. Similarly inappropriate causal language is used to explain the links between emotion and cognition, on which the authors are only speculating about directionality.

For Part 2 of the analysis (assessing an interaction), I would like to see more documentation of the model used, particularly all variables and coefficients, as well as error.

I would like to know how the authors dealt with the fact that they had categorical data. How did the SEM accommodate this?

2. Has the statistical analysis been performed appropriately and rigorously? I don’t know.

I don’t know, as I would need more information from the authors to assess (see comments above).

Other:

I see the dependent variable of support for climate policy as questionable. Support for policy is typically influenced by many factors not measured in this study, such as political preferences and worldview, especially when it comes to climate change. To say that the SEM in this study predicts climate policy is likely obscuring important predictive variables related to one’s political position. The lack of including a measure for political views undercuts the relationship between art viewing and climate policy support suggested by this study.

6. PLOS authors have the option to publish the peer review history of their article (what does this mean?). If published, this will include your full peer review and any attached files.

Reviewer #1: No

---

## [Author Response · Author response to Decision Letter 0]

8 Jan 2021

Please see the attached document with responses to the reviewer and the editor.

---

## [Decision Letter · Decision Letter 1]

26 Jan 2021

PONE-D-20-32925R1

Visual art inspired by climate change – An analysis of audience reactions to 37 artworks presented during 21st UN climate summit in Paris

PLOS ONE

Dear Dr. Klöckner,

Thank you for submitting your manuscript to PLOS ONE. After careful consideration, we feel that it has merit but does not fully meet PLOS ONE’s publication criteria as it currently stands. Therefore, we invite you to submit a revised version of the manuscript that addresses the points raised during the review process.

We look forward to receiving your revised manuscript.

Kind regards,

Valerio Capraro

Academic Editor

PLOS ONE

Additional Editor Comments (if provided):

The reviewer thinks that the paper has been improved, but still suggests a minor revision before publication. Please address these last comments at your earliest convenience. I am looking forward for the final version.

Reviewers' comments:

Reviewer's Responses to Questions

**Comments to the Author**

1. If the authors have adequately addressed your comments raised in a previous round of review and you feel that this manuscript is now acceptable for publication, you may indicate that here to bypass the “Comments to the Author” section, enter your conflict of interest statement in the “Confidential to Editor” section, and submit your "Accept" recommendation.

Reviewer #1: (No Response)

2. Is the manuscript technically sound, and do the data support the conclusions?

Reviewer #1: Yes

3. Has the statistical analysis been performed appropriately and rigorously? 

Reviewer #1: Yes

4. Have the authors made all data underlying the findings in their manuscript fully available?

Reviewer #1: Yes

5. Is the manuscript presented in an intelligible fashion and written in standard English?

Reviewer #1: Yes

6. Review Comments to the Author

Reviewer #1: Thank you for taking such care in revising this manuscript. It is much clearer all around. I appreciate the new, detailed explanations of methods and the tempering of claims. The addition in the Limitations section is appreciated. The authors have responded to all of my feedback well. I have two remaining considerations:

First, I am slightly concerned about the unreliability of the Identifying with the Artist construct. While I agree with the authors that the unreliable multi-item variable is better than just a single item, my bigger question is what happens to the model when you drop the construct entirely? I recognize that you want to keep it in, as it aligns with your theory, but if it's unreliable, is it actually useful? I think the authors need to address this in some way--either further justify your keeping it, or show results with that construct dropped.

Second, very minorly, there are a few typos. Additionally, the new table (Table 4) could use some extra clarification about the Thresholds, as described in the authors' response to reviewers.

7. PLOS authors have the option to publish the peer review history of their article (what does this mean?). If published, this will include your full peer review and any attached files.

Reviewer #1: No

---

## [Author Response · Author response to Decision Letter 1]

4 Feb 2021

Please see the attached document with responses to the reviewer.

---

## [Editor Report · Decision Letter 2]

5 Feb 2021

Visual art inspired by climate change – An analysis of audience reactions to 37 artworks presented during 21st UN climate summit in Paris

PONE-D-20-32925R2

Dear Dr. Klöckner,

We’re pleased to inform you that your manuscript has been judged scientifically suitable for publication and will be formally accepted for publication once it meets all outstanding technical requirements.

Kind regards,

Valerio Capraro

Academic Editor

PLOS ONE
---

## [Editor Report · Acceptance letter]

9 Feb 2021

PONE-D-20-32925R2 

Visual art inspired by climate change – An analysis of audience reactions to 37 artworks presented during 21st UN climate summit in Paris 

Dear Dr. Klöckner:

I'm pleased to inform you that your manuscript has been deemed suitable for publication in PLOS ONE. Congratulations! Your manuscript is now with our production department. 

Kind regards, 

on behalf of

Dr. Valerio Capraro 

Academic Editor

PLOS ONE